# (PASS) Visual Prompt Locates Good Structure Sparsity through a Recurrent HyperNetwork

## Abstract

Large-scale neural networks have demonstrated remarkable performance in different domains like vision and language processing, although at the cost of massive computation resources. As illustrated by compression literature, structural model pruning is a prominent algorithm to encourage model efficiency, thanks to its acceleration-friendly sparsity patterns. One of the key questions of structural pruning is how to estimate the channel significance. In parallel, work on data-centric AI has shown that prompting-based techniques enable impressive generalization of large language models across diverse downstream tasks. In this paper, we investigate a charming possibility - *leveraging visual prompts to capture the channel importance and derive high-quality structural sparsity*. To this end, we propose a novel algorithmic framework, namely PASS. It is a tailored hypernetwork to take both visual prompts and network weight statistics as input, and output layer-wise channel sparsity in a recurrent manner. Such designs consider the intrinsic channel dependency between layers. Comprehensive experiments across multiple network architectures and six datasets demonstrate the superiority of PASS in locating good structural sparsity. For example, at the same FLOPs level, PASS subnetworks achieve $1\% \sim 3\%$ better accuracy on Food101 dataset; or with a similar performance of $80\%$ accuracy, PASS subnetworks obtain $0.35\times$ more speedup than the baselines. Codes are provided in the supplements.

## 1 Introduction

Recently, large-scale neural networks, particularly in the field of vision and language modeling, have received upsurging interest due to the promising performance for both natural language (Brown et al., 2020; Chiang et al., 2023; Touvron et al., 2023) and vision tasks (Dehghani et al., 2023; Bai et al., 2023). While these models have delivered remarkable performance, their colossal model size, coupled with their vast memory and computational requirements, pose significant obstacles to model deployment. To solve this daunting challenge, model compression techniques have re-gained numerous attention (Dettmers et al., 2022; Xiao et al., 2023; Ma et al., 2023; Frantar & Alistarh, 2023; Sun et al., 2023; Jaiswal et al., 2023) and knowledge distillation can be further adopted on top of them to recover optimal performance (Huang et al., 2023; Sun et al., 2019; Kim et al., 2019). Among them, model pruning is a well-established method known for its capacity to reduce model size without compromising performance (LeCun et al., 1989; Han et al., 2015; Molchanov et al., 2016) and structural model pruning has garnered significant interest due to its ability to systematically eliminate superfluous structural components, such as entire neurons, channels, or filters, rather than individual weights, making it more hardware-friendly (Li et al., 2017; Liu et al., 2017a; Fang et al., 2023; Yin et al., 2023).

In the context of structural pruning for *vision models*, the paramount task is the estimation of the importance of each structure component, such as channel or filters. It is a fundamental challenge since it requires dissecting the neural network behavior and a precise evaluation of the relevance of individual structural sub-modules. Previous methodologies (Liu et al., 2017b; Fang et al., 2023; Wang et al., 2021; Murti et al., 2022; Nonnenmacher et al., 2022) have either employed heuristics or developed learning pipelines to derive scores, achieving notable performance. Recently, the prevailingness of natural language prompts (Ouyang et al., 2022; Ganguli et al., 2023) has facilitated an emerging wisdom that the success of AI is deeply rooted in the quality and specificity of data

that is originally created by human (Zha et al., 2023; Gunasekar et al., 2023). Techniques such as in-context learning (Chen et al., 2022a; Wei et al., 2022; Min et al., 2022) and prompting (Razdaibiedina et al., 2023; Dong et al., 2022; Chen et al., 2023; Liu et al., 2021c; Chen et al., 2022b) have been developed to create meticulously designed prompts or input templates to escalate the output quality of LLMs. These strategies bolster the capabilities of LLMs and consistently achieve notable success across diverse downstream tasks. This offers a brand new angle for addressing the intricacies of structural pruning on importance estimation of vision models: *How can we leverage the potentials within the input space to facilitate the dissection of the relevance of each individual structural component across layers, thereby enhancing structural sparsity?*

One straightforward approach is directly editing input through visual prompt (Jia et al., 2022) to enhance the performance of compressed vision models (Xu et al., 2023). The performance upper bound of this approach largely hinges on the quality of the sparse model achieved by pruning, given that prompt learning is applied post-pruning. Moreover, when pruning is employed to address the intricate relevance between structural components across layers, the potential advantages of using visual prompts are not taken into consideration.

Therefore, we posit that probing judicious input editing is imperative for structural pruning to examine the importance of structural components in vision models. The **crux of our research** lies in embracing an innovative **_data-centric_** viewpoint towards structural pruning. Instead of designing or learning prompts on top of compressed models, we develop a novel end-to-end framework for channel pruning, which identifies and retains the most crucial channels across models by incorporating visual prompts, referred to as `PASS`.

Moreover, the complexities associated with inherent channel dependencies render the generation of sparse channel masks a challenging task. Due to this reason, many previous arts of pruning design delicate pruning metrics to recognize sparse subnetworks with smooth gradient flow (Wang et al., 2020; Evci et al., 2022; Pham et al., 2022). To better handle the channel dependencies across layers during channel pruning, we propose to learn sparse masks using a **recurrent mechanism**. Specifically, the learned sparse mask for the recent layer largely depends on the mask from the previous layer in an efficient recurrent manner, and all the masks are learned by incorporating the extra information provided by visual prompts. The `PASS` framework is shown in Figure 1. Our contributions are summarized as follows:

- We probe and comprehend the role of the input editing in the context of channel pruning, and confirming the imperative to integrate visual prompts for crucial channel discovery.
- To handle the complex dependence caused by channel elimination across layers, we further develop a recurrent mechanism to efficiently learn layer-wise sparse masks by taking both the sparse masks from previous layers and visual prompts into consideration. Anchored by these innovations, we propose `PASS`, a pioneering framework dedicated to proficient channel pruning in convolution neural networks from a data-centric perspective.
- Through comprehensive evaluations across six datasets containing {CIFAR-10, CIFAR-100, Tiny-ImageNet, Food101, DTD, StanfordCars} and four architectures including {ResNet-18, ResNet-34, ResNet-50, VGG}, our results consistently demonstrate `PASS`'s significant potential in enhancing both the performance of the resultant sparse models and computational efficiency.
- More interestingly, our empirical studies reveal that the sparse channel masks and the hypernetwork produced by `PASS` exhibit superior transferability, proving beneficial for a range of subsequent tasks.

## 2 RELATED WORK

**Structural Network Pruning.** Structural pruning achieves network compression through entirely eliminating certain superfluous components from the dense network. In general, structural pruning follows three steps: (i) pre-training a large, dense model; (ii) pruning the unimportant channels based on criteria, and (iii) finetuning the pruned model to recover optimal performance. The primary contribution of various pruning approaches is located in the second step: proposing proper pruning metrics to identify the importance of channels. Some commonly-used pruning metric includes but not limited to weight norm (Li et al., 2016; He et al., 2018a; Yang et al., 2018), Taylor

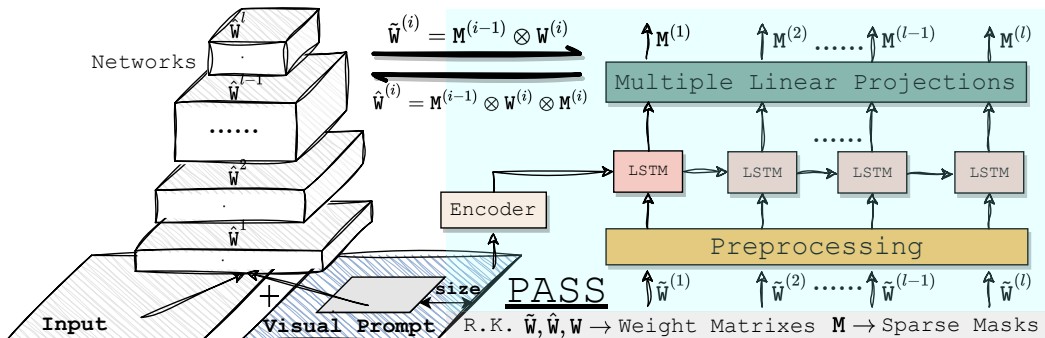

Figure 1: The overall framework of PASS. (*Left*) Our pruning target is a convolutional neural network (CNN) that takes images and visual prompts as input. (*Right*) The PASS hyper-network integrates the information from visual prompts and layer-wise weight statistics, then determines the significant structural topologies in a recurrent fashion.

expansion Molchanov et al. (2016; 2019), feature-maps reconstruction error (He et al., 2018b; 2017; Luo et al., 2017; Zhuang et al., 2018), feature-maps rank (Lin et al., 2020a), KL-divergence Luo & Wu (2020), greedy forward selection with largest loss reduction Ye et al. (2020), feature-maps discriminant information Hou & Kung (2020b;a); Kung & Hou (2020).

**Prompting.** In the realm of natural language processing, prompting has been acknowledged as an effective strategy to adapt pre-trained models to specific tasks (Liu et al., 2023a). The power of this technique was highlighted by GPT-3's successful generalization in transfer learning tasks using carefully curated text prompts (Brown et al., 2020). Researchers have focused on refining text prompting methods (Shin et al., 2020), (Jiang et al., 2020) and developed a technique known as Prompt Tuning. This approach involves using prompts as task-specific continuous vectors optimized during fine-tuning (Li & Liang, 2021), (Lester et al., 2021), (Liu et al., 2021c), offering comparable performance to full fine-tuning with a significant reduction in parameter storage and optimization. Prompt tuning's application in the visual domain has seen significant advancement recently. Pioneered by Bahng et al. (2022), who introduced prompt parameters to input images, the concept was expanded by Chen et al. (2023) to envelop input images with prompt parameters. Jia et al. (2022) took this further, proposing visual prompt tuning for Vision Transformer models. Subsequently, Liu et al. (2023b); Zheng et al. (2022); Zhang et al. (2022) designed a prompt adapter to enhance these prompts. Concurrently, Zang et al. (2022); Zhou et al. (2022b;a) integrated visual and text prompts in vision-language models, boosting downstream performance.

**Hypernetwork.** Hypernetworks represent a specialized form of network architecture, specifically designed to generate the weights of another Deep Neural Network (DNN). This design provides a meta-learning approach that enables dynamic weight generation and adaptability, which is crucial in scenarios where flexibility and learning efficiency are paramount. Initial iterations of hypernetworks, as proposed by Zhang et al. (2018); Galanti & Wolf (2020); David et al. (2016); Li et al. (2020), were configured to generate the weights for an entire target DNN. While this approach is favorable for smaller and less complex networks, it constrains the efficacy of hypernetworks when applied to larger and more intricate ones. To address this limitation, subsequent advancements in hypernetworks have been introduced, such as the component-wise generation of weights (Zhao et al., 2020; Alaluf et al., 2022; Mahabadi et al., 2021) and chunk-wise generation of weights (Chauhan et al., 2023). Diverging from the initial goal of hypernetworks, our work employs them to fuse visual prompts and model information for generating sparse channel masks.

## 3 PASS: VISUAL PROMPT LOCATES GOOD STRUCTURE SPARSITY

**Notations.** Let us consider a CNN with $l$ layers, and each layer $i$ contains its corresponding weight tensor $W^{(i)} \in \mathbb{R}^{C_O^i \times C_I^i \times K^i \times K^i}$, where $\{C_O^i, C_I^i, \text{and } K^i\}$ are the number of output/input channels and convolutional kernal size, respectively. The entire parameter space for the network is defined as $W = \{W^{(i)}\}|_{i=1}^{l}$. Similarly, a layer-wise binary mask is represented by $M^{(i)}$, where "0"/"1" indicates removing/maintaining the associated channel. $V$ denotes our visual prompts. $(x, y) \in \mathcal{D}$ denotes the data of a target task.

**Rationale.** In the realm of structural pruning for deep neural networks, one of the key challenges is how to derive channel-wise importance scores for each layer. Conventional mechanisms estimate the channel significance either in a global or layer-wise manner (He et al., 2017; Li et al., 2016; Fang et al., 2023; Zhu & Gupta, 2017), neglecting the sequential dependency between adjacency layers. Meanwhile, the majority of prevalent pruning methods are designed in a *model-centric* fashion (Fang et al., 2023; Li et al., 2016; Lin et al., 2021; 2022; Liu et al., 2017b; Wang et al., 2021). In contrast, an ideal solution to infer the high-quality sparse mask for one neural network layer $i$ should satisfy several conditions as follows:

① $\texttt{M}^{(i)}$ *should be dependent to* $\texttt{M}^{(i-1)}$. The sequential dependency between layers should be explicitly considered. It plays an essential role in encouraging gradient flow throughout the model (Wang et al., 2020; Pham et al., 2022), by preserving structural "pathways".

② $\texttt{M}^{(i)}$ *should be dependent to* $\texttt{W}^{(i)}$. The statistics of network weights are commonly appreciated as powerful features for estimating channel importance (Liu et al., 2017b; Li et al., 2016).

③ $\texttt{M}^{(i)}$ *should be dependent to* $\texttt{V}$. Motivated by the *data-centric* advances in NLP, such prompting can contribute to the dissecting and understanding of model behaviors (Razdaibiedina et al., 2023; Dong et al., 2022; Chen et al., 2023; Liu et al., 2021c; Chen et al., 2022b).

Therefore, it can be expressed as $\texttt{M}^{(i)} = f(\texttt{M}^{(i-1)}, \texttt{W}^{(i)}, \texttt{V})$, where the generation of a channel mask for layer $i$ depends on the weights in the current layer, the previous layers' mask, and visual prompts.

### 3.1 INNOVATIVE DATA-MODEL CO-DESIGNS THROUGH A RECURRENT HYPERNETWORK

To meet the aforementioned requirements, $\texttt{PASS}$ is proposed as illustrated in Figure 1, which enables the data-model co-design pruning via a recurrent hyper-network. Details are presented below.

**Modeling the Layer Sequential Dependency.** The recurrent hyper-network in $\texttt{PASS}$ adopts a Long Short-Term Memory (LSTM) backbone since it is particularly suitable for capturing sequential dependency. It enables an "auto-regressive" way to infer the structural sparse mask. To be specific, the LSTM mainly utilizes the previous layer's mask $\texttt{M}^{(i-1)}$, the current layer's weights $\texttt{W}^{(i)}$, and a visual prompt $\texttt{V}$ as follows:

$$\texttt{M}^{(i)} = \texttt{LSTM}_\theta(\widetilde{\texttt{W}}^i, g_\omega(\texttt{V})) \,, \widetilde{\texttt{W}}^{(i)} = \texttt{M}^{(i-1)} \otimes \texttt{W}^{(i)}, \; \texttt{M}^{(0)} = \texttt{LSTM}(\texttt{W}^{(i)}, g_\omega(\texttt{V})), \qquad (1)$$

where the visual prompt $\texttt{V}$ provides an initial hidden state for the LSTM hyper-network, $\theta$ is the parameters of the LSTM model, and $g_\omega(\texttt{V})$ is the extra encoder to map the visual prompt into an embedding space. The channel-wise sparse masks ($\texttt{M}^{(i)}$) generated from the hyper-network are utilized to prune the weights of each layer as expressed by $\widehat{\texttt{W}}^{(i)} = \texttt{M}^{(i-1)} \otimes \texttt{W}^{(i)} \otimes \texttt{M}^{(i)}$. $\texttt{M}^{(i-1)} \otimes \texttt{W}^{(i)}$ represents the pruning of in-channels while $\texttt{W}^{(i)} \otimes \texttt{M}^{(i)}$ denotes the pruning of out-channels.

**Visual Prompt Encoder.** An encoder is used to extract representations from the raw visual prompt $\texttt{V}$. $g_\omega(\texttt{V})$ denotes a three-layer convolution network and $\omega$ are the parameters for the CNN $g_\omega(\cdot)$. The dimension of extracted representations equals the dimension of the hidden state of the LSTM model. A learnable embedding will serve as the initial hidden state for the LSTM model.

**Preprocessing the Weight.** The in-channel pruned weights $\widetilde{\texttt{W}}^{(i)}$ is a 4D matrix. In order to take this weight information, it is first transformed into a vector of length equal to the number of out-channels by averaging the weights over the $C_I^i \times K^i \times K^i$ dimensions. Then, these vectors are padded by zero elements to unify their length.

**Converting Embedding to Channel-wise Sparse Mask.** Generating layer-wise channel masks from the LSTM module presents two challenges: (1) it outputs embeddings of a uniform length, whereas the number of channels differs at each layer; (2) producing differentiable channel masks directly from this module is infeasible. To tackle these issues, $\texttt{PASS}$ adopts a two-step approach: ① An independent linear layer is employed to map the learned embeddings onto channel-wise important scores corresponding to each layer. ② During the forward pass in training, the binary channel mask $\texttt{M}$ is produced by setting the $(1 - s) \times 100\%$ elements with the highest channel-wise important scores to 1, with the rest elements set to 0. In the backward pass, it is optimized by leveraging the

straight-through estimation method (Bengio et al., 2013). Here the $s \in (0, 1)$ denotes the channel sparsity of the network layer.

For achieving an optimal non-uniform layer-wise sparsity ratio, we adopt global pruning (Huang et al., 2022) that eliminates the channels associated with the lowest score values from all layers during each optimization step. This approach is grounded in the findings of Huang et al. (2022); Liu et al. (2021b); Fang et al. (2023), which demonstrate that layer-wise sparsity derived using this method surpasses other extensively researched sparsity ratios.

## 3.2 How to Optimize the Hypernetwork in PASS

**Learning PASS.** The procedures of learning PASS involves a jointly optimization of the visual prompt V, encoder weights $\omega$, and LSTM's model weights $\theta$. Formally, it can be described below:

$$\min_{\theta, \omega, \mathtt{V}} \mathcal{L}(\Phi_{\widehat{\mathtt{W}}}(\boldsymbol{x} + \mathtt{V}), y), \ \widehat{\mathtt{W}}^{(i)} = \mathtt{M}^{(i-1)} \otimes \mathtt{W}^{(i)} \otimes \mathtt{M}^{(i)}, \tag{2}$$

Where $\Phi_{\widehat{\mathtt{W}}}(\cdot)$ is the target CNN with weights $\widehat{\mathtt{W}}$, $\boldsymbol{x}$ and $y$ are the input image and its groundtruth label. Note that $\mathtt{M}^{(i)}$ is generated by $\mathtt{LSTM}_\theta(\widetilde{\mathtt{W}}^i, g_\omega(\mathtt{V}))$ as described in Equation 1. The objective of this learning phase is to optimize the PASS model to generate layer-wise channel masks, leveraging both a visual prompt V and the inherent model weight statistics as guidance. After that, the obtained sparse subnetwork will be further fine-tuned on the downstream dataset.

**Fine-tuning Sparse Subnetwork.** The procedures of subnetwork fine-tuning involve the optimization of the visual prompt V and model weights W, which can be expressed by:

$$\min_{\mathtt{W}, \mathtt{V}} \mathcal{L}(\Phi_{\widehat{\mathtt{W}}}(\boldsymbol{x} + \mathtt{V}), y), \tag{3}$$

where $\widehat{\mathtt{W}} = \mathtt{M}^{(i-1)} \otimes \mathtt{W}^{(i)} \otimes \mathtt{M}^{(i)}$ and the sparse channel mask M is fixed.

## 4 Experiments

In this section, we empirically demonstrate the effectiveness of our proposed PASS method against various baselines across multiple datasets and models. Additionally, we evaluate the transferability of the sparse channel masks and the hypernetwork learned by PASS. Further, we validate the superiority of our specific design by a series of ablations studies.

To evaluate PASS, we follow the widely-used evaluation of visual prompting which is pre-trained on large datasets and evaluated on various target domains (Chen et al., 2023; Jia et al., 2022). Specifically, this process is accomplished by two steps: (1) Identifying an optimal structural sparse neural network based on a pre-trained model and (2) Fine-tuning the structural sparse neural network on the target task. During the training process, we utilize the Frequency-based Label Mapping $FLM$ as presented by Chen et al. (2023) to facilitate the mapping of the logits from the pre-trained model to the logits of the target tasks.

### 4.1 Implementation Setups

**Architectures and Datasets.** We evaluate PASS using four pre-trained models: ResNet-18, ResNet-34, ResNet-50 (He et al., 2016), and VGG-16 without BatchNorm2D (Simonyan & Zisserman, 2014), all pre-trained on ImageNet-1K (Deng et al., 2009). Our evaluation contains six target tasks: Tiny-ImageNet (Deng et al., 2009), CIFAR-10/100 (Krizhevsky et al., 2009), DTD (Cimpoi et al., 2014), StanfordCars (Krause et al., 2013), and Food101 (Bossard et al., 2014). The size of the inputs is scaled to $224 \times 224$ during our experiments.

**Baselines.** We select five popular structural pruning methods as our baselines: (1) *Group-L1 structural pruning* (Li et al., 2017; Fang et al., 2023) reduces the network channels via $l_1$ regularization. (2) *GrowReg* (Wang et al., 2021) prunes the network channels via $l_2$ regularization with a growing penalty scheme. (3) *Slim* (Liu et al., 2017b) imposes channel sparsity by applying $l_1$ regularization to the scaling factors in batch normalization layers. (4) *DepGraph* (Fang et al., 2023) models the inter-layer dependency and group-coupled parameters for pruning and (5) *ABC Pruner* (Lin et al., 2020b) performs channel pruning through automatic structure search.

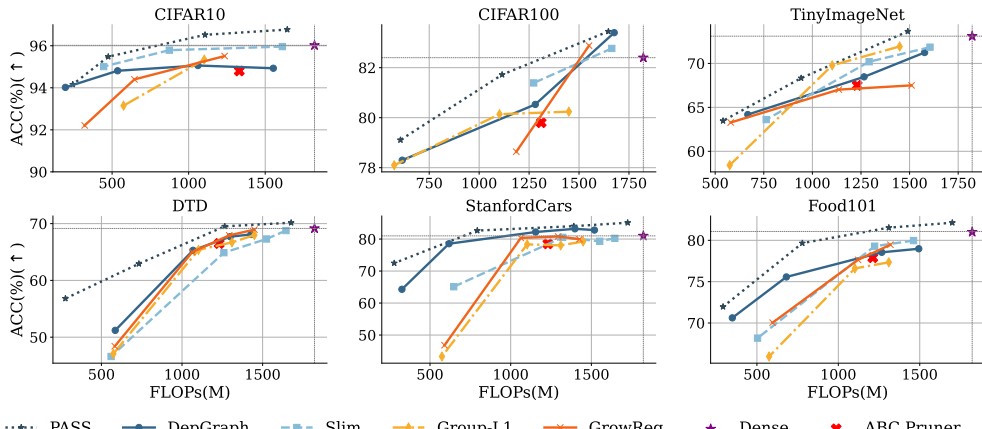

Figure 2: Test accuracy of channel-pruned networks across multiple downstream tasks based on the pre-trained ResNet-18 model.

**Training and Evaluation.** We utilize off-the-shelf models from Torchvision [1] as the pre-trained models. During the pruning phase, we employ the SGD optimizer for the visual prompt, while the AdamW optimizer is used for the visual prompt encoder and the LSTM model for generating channel masks. Regarding the baselines, namely Group-L1 structural pruning, GrowReg, Slim, and Dep-Graph, they are trained based on this implementation [2] and ABC Prunner is trained based on their official public code [3]. During the fine-tuning phase, all pruned models, inclusive of those from PASS and the aforementioned baselines, are fine-tuned with the same hyper-parameters. We summarize the implementation details and hyper-parameters for PASS in Appendix B. For all experiments, we report the accuracy of the downstream task during testing and the floating point operations (FLOPs) for measuring the efficiency.

## 4.2 PASS FINDS GOOD STRUCTURAL SPARSITY

In this section, we first validate the effectiveness of PASS across multiple downstream tasks and various model architectures. Subsequently, we investigate the transferability of both the generated channel masks and the associated model responsible for generating them.

**Superior Performance across Downstream Tasks**. In Figure 2, we present the test accuracy of the PASS method in comparison with several baseline techniques, including Group-L1, GrowReg, DepGraph, Slim, and ABC Prunner. The evaluation includes six downstream tasks: CIFAR-10, CIFAR-100, Tiny-ImageNet, DTD, StanfordCars, and Food101. The accuracies are reported against varying FLOPs to provide a comprehensive understanding of PASS's efficiency and performance.

From Figure 2, several salient observations can be drawn: ❶ PASS consistently demonstrates superior accuracy across varying FLOPs values for all six evaluated downstream tasks. On one hand, PASS achieves higher accuracy under the same FLOPs. For example, it achieves $1\% \sim 3\%$ higher accuracy than baselines under 1000M FLOPs among all the datasets. On the other hand, PASS attains higher speedup[4] in achieving comparable accuracy levels. For instance, to reach accuracy levels of $96\%$, $81\%$, and $80\%$ on CIFAR10, StanfordCars, and Food101 respectively, the PASS method consistently realizes a speedup of at least $0.35\times$ (900 VS 1400), outperforming the most competitive baseline. This consistent performance highlights the robustness and versatility of the PASS method across diverse scenarios. ❷ In terms of resilience to pruning, PASS exhibits a more gradual reduction in accuracy as FLOPs decrease. This trend is notably more favorable when compared with the sharper declines observed in other baseline methods. ❸ Remarkably, at the higher FLOPs levels, PASS not only attains peak accuracies but also surpasses the performance metrics of

---

[1] https://pytorch.org/vision/stable/index.html

[2] https://github.com/VainF/Torch-Pruning

[3] https://github.com/lmbxmu/ABCPruner

[4] Following Fang et al. (2023), we report the theoretical speedup ratios and it is defined as $\frac{\text{FLOPs}_{\text{PASS}} - \text{FLOPs}_{\text{baseline}}}{\text{FLOPs}_{\text{baseline}}}$

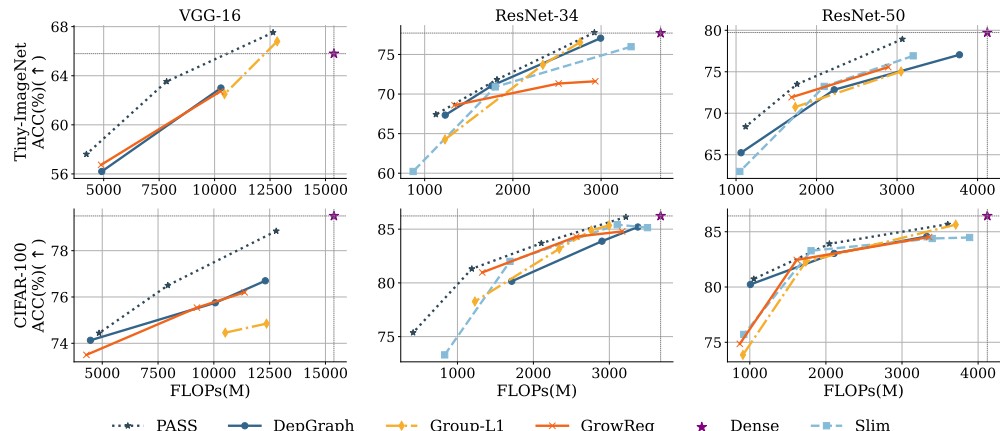

Figure 3: Test accuracy of channel-pruned networks across various architectures based on CIFAR-100 and Tiny-ImageNet datasets.

Table 1: Transferability: Applying Channel Masks and Hypernetworks Learned from Tiny-ImageNet to CIFAR-100 and StanfordCars. The gray color denotes our method.

| Channel Sparsity | 10% | | 30% | | 50% | |
|---|---|---|---|---|---|---|
| | StanfordCars | CIFAR-100 | StanfordCars | CIFAR-100 | StanfordCars | CIFAR-100 |
| DepGraph | 75.79 | 81.60 | 69.26 | 76.90 | 45.10 | 69.40 |
| Slim | 58.10 | 80.27 | 43.00 | 71.86 | 26.3 | 68.56 |
| Group-L1 | 76.50 | 79.80 | 58.30 | 72.60 | 20.40 | 58.50 |
| Growreg | 70.60 | 80.79 | 50.30 | 72.27 | 41.80 | 65.80 |
| Transfer Channel Mask | 83.50 | 82.45 | 79.70 | 80.83 | 76.60 | 78.81 |
| Hypernetwork | 84.31 | 82.49 | 79.88 | 80.98 | 76.80 | 78.67 |

the fully fine-tuned dense models. For instance, `PASS` excels the fully fine-tuned dense models with {1.05%, 0.99%, 1.06%} on CIFRAR100, DTD and FOOD101 datasets.

**Superior Performance across Model Architectures.** We further evaluate the performance of `PASS` across multiple model architectures, namely VGG-16 without batch normalization [5], ResNet-34, and ResNet-50 and compare it with the baselines. The results are shown in 3. We observe that our `PASS` achieves a competitive performance across all architectures, often achieving accuracy close to or even surpassing the dense models while being more computationally efficient. For instance, To achieve an accuracy of 75% on Tiny-ImageNet using ResNet-34/ResNet-50 and 66% accuracy using VGG-16, our `PASS` requires 0% ∼ 12% fewer FLOPs compared to the most efficient baseline. These observations suggest that `PASS` can effectively generalize across different architectures, maintaining a balance between computational efficiency and model performance.

## 4.3 TRANSFERABILITY OF LEARNED SPARSE STRUCTURE

Inspired by studies suggesting the transferability of subnetworks between tasks (Chen et al., 2020; 2021). We investigate the transferability of `PASS` by posing two questions:(1) *Can the sparse channel masks, learned in one task, be effectively transferred to other tasks?* (2) *Is the hypernetwork, once trained, applicable to other tasks?* To answer Question (1), we test the accuracy of subnetworks found on Tiny-ImageNet when fine-tuning on CIFAR-10 and CIFAR-100 and a pre-trained ResNet-18. To answer Question (2), we measure the accuracy of the subnetwork finetuning on the target datasets, i.e., CIFAR-10 and CIFAR-100. This subnetwork is obtained by applying hypernetworks, trained on Tiny-ImageNet, to the visual prompts of the respective target tasks. The results are reported in Table 1. We observe that the channel mask and the hypernetwork, both learned by `PASS`, exhibit significant transferability on target datasets, highlighting their benefits across various subsequent tasks. More interestingly, the hypernetwork outperforms transferring the channel mask in most target tasks, providing two hints: ❶ Our learned hypernetworks can sufficiently capture the

---

[5]The baseline Slim (Liu et al., 2017b) is not applicable to this architecture.

Table 2: Ablations for `PASS` based on CIFAR-100 using a pre-trained ResNet-18.

| Channel Sparsity = | | 10% | 30% | 50% | 70% |
|---|---|---|---|---|---|
| | LSTM+VP | 82.66 | 81.20 | 77.94 | 72.01 |
| Input Ablations | LSTM+Weights | 82.83 | 81.13 | 77.83 | 72.45 |
| | LSTM+Weights+VP(Ours) | 83.45 | 81.72 | 79.11 | 73.53 |
| | ConVNet+VP | 83.21 | 81.09 | 78.15 | 72.31 |
| Architecture Ablations | MLP+VP+Weights | 83.23 | 81.07 | 77.84 | 72.38 |
| | LSTM+Weights+VP(Ours) | 83.45 | 81.72 | 79.11 | 73.53 |

important topologies in downstream networks. Note that there is no parameter tuning for the hyper-networks and only with an adapted visual prompt. ❷ The visual prompt can effectively summarize the topological information from downstream neural networks, enabling superior sparsification.

# 5 ABLATIONS AND EXTRA INVESITIGATIONS

## 5.1 ABLATIONS ON `PASS`

To evaluate the effectiveness of `PASS`, we pose two interesting questions about the design of its components: (1) *how do visual prompts and model weights contribute?* (2) *is the recurrent mechanism crucial for mask finding?* To answer the above questions, we conduct a series of ablation studies utilizing a pre-trained ResNet-18 on CIFAR-100. The extensive investigations contain (1) *dropping either the visual prompt or model weights*; (2) *destroy the recurrent nature in our hypernetwork*, such as using a Convolutional Neural Network (CNN) or a Multilayer Perceptron (MLP) to replace LSTM. The results are collected in Table 2. We observe that ❶ The exclusion of either the visual prompt or model weights leads to a pronounced drop in test accuracy (e.g., $83.45\% \to 82.83\%$ and $82.66\%$ respectively at $90\%$ channel density), indicating the essential interplay role of both visual prompt and model weights in sparsification. ❷ If the recurrent nature in our design is destroyed, *i.e.*, MLP or CNN methods variants, it suffers a performance decrement (*e.g.*, $81.72\% \to 81.07\%$ and $81.09\%$ respectively at $70\%$ channel density). It implies a Markov property during the sparsification of two adjacent layers, which echoes the sparsity pathway findings in Wang et al. (2020).

## 5.2 ABLATIONS ON VISUAL PROMPT

A visual prompt is a patch integrated with the input, as depicted in Figure 1. Two prevalent methods for incorporating the visual prompt into the input have been identified in the literature (Chen et al., 2023; Bahng et al., 2022):(1) Adding to the input (abbreviated as **"Additive visual prompt"**. (2)Expanding around the perimeter of the input, namely, the input is embedded into the central hollow section of the visual prompt (abbreviated as **"Expansive visual prompt"**). As discussed in section 5.1, visual prompt (VP) plays a key role in `PASS`. Therefore, we pose such a question:*How do the strategies and size of VP influence the performance of `PASS`?* To address this concern, we conduct experiments with "Additive visual prompt" and "Expansive visual prompt" respectively on CIFAR-100 using a pre-trained ResNet-18 under $10\%$, $30\%$ and $50\%$ channel sparsities, and we also show the performance of `PASS` with varying the VP size from 0 to 48. The results are shown in Figure 4. We conclude that ❶ "Additive visual prompt" performs better than "Expansive visual prompt" across different sparsities. The disparity might be from the fact that "Expansive visual prompt" requires resizing the input to a smaller dimension, potentially leading to information loss, a problem that "Additive visual prompt" does not face. ❷ The size of VP impacts the performance of `PASS`. We observe that test accuracy initially rises with the increase in VP size but starts to decline after reaching a peak at size 16. A potential explanation for this decline is that the larger additive VP might overlap a significant portion of the input, leading to the loss of crucial information.

## 5.3 IMPACT OF HIDDEN SIZE IN HYPERNETWORK

It is well-known that model size is an important factor impacting its performance, inducing the question *how does the size of hypernetwork influence the performance of PASS?*. To address this concern, we explore the impact of the hypernetwork hidden sizes on `PASS` by varying the hidden size of the proposed hypernetwork from 32 to 256 and evaluate its performance on CIFAR100 using

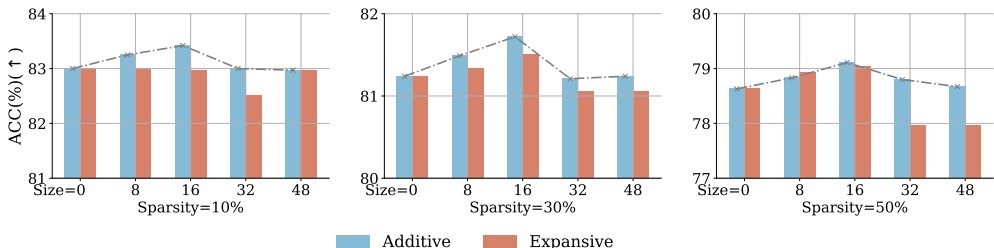

Figure 4: Ablation study on visual prompt strategies and their sizes. Experiments are conducted on CIFAR-100 and a pre-trained ResNet-18.

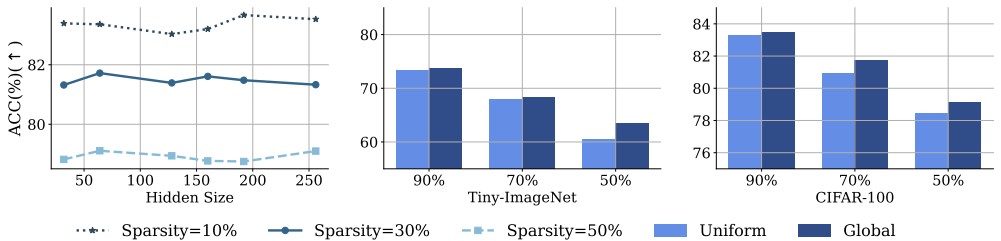

Figure 5: (1)Ablation study of the hypernetwork's hidden size (Left Figure) using a pre-trained ResNet-18 on CIFAR-100. (2)Comparison between Global Pruning and Uniform Pruning strategies (Middle and Right Figures) using a pre-trained ResNet-18 on CIFAR-100 and Tiny-Imagenet.

a pre-trained ResNet-18 model under $10\%$, $30\%$ and $50\%$ channel sparsity respectively. The results are presented in the left figure of Figure 5. We observe that the hidden size of the hypernetwork doesn't drastically affect the accuracy. While there are fluctuations, they are within a small range, suggesting that the hidden size is not a dominant factor in influencing the performance of PASS.

## 5.4 UNIFORM PRUNING VS GLOBAL PRUNING

When converting the channel-wise importance scores into the channel masks, there are two prevalent strategies: (1) *Uniform Pruning.* (Ramanujan et al., 2020; Huang et al., 2022) It prunes the channels of each layer with the lowest important scores by the same proportion. (2) *Global Pruning.* (Huang et al., 2022; Fang et al., 2023) It prunes channels with the lowest important scores from all layers, leading to varied sparsity across layers. In this section, we evaluate the performance of global pruning and uniform pruning for PASS on CIFAR-100 using a pre-trained ResNet-18, with results presented in Figure 5. We observe that global pruning consistently yields higher test accuracy than uniform pruning, indicating its superior suitability for PASS, also reconfirming the importance of layer sparsity in sparsifying neural networks (Liu et al., 2022; Huang et al., 2022). For a detailed overview of the sparsity learned at each layer using global pruning, please refer to Appendix C.

## 6 CONCLUSION

In this paper, we delve deep into structural model pruning, with a particular focus on leveraging the potential of visual prompts for discerning channel importance in vision models. Our exploration highlights the key role of the input space and how judicious input editing can significantly influence the efficacy of structural pruning. We propose PASS, an innovative, end-to-end framework that harmoniously integrates visual prompts, providing a data-centric lens to channel pruning. Our recurrent mechanism adeptly addressed the intricate channel dependencies across layers, ensuring the derivation of high-quality structural sparsity.

Extensive evaluations across six datasets and four architectures underscore the prowess of PASS. The PASS framework excels not only in performance and computational efficiency but also demonstrates that its pruned models possess notable transferability. In essence, this research paves a new path for channel pruning, underscoring the importance of intertwining data-centric approaches with traditional model-centric methodologies. The fusion of these paradigms, as demonstrated by our findings, holds immense promise for the future of efficient neural network design.

## 7 REPRODUCIBILITY STATEMENT

The authors have made an extensive effort to ensure the reproducibility of algorithms and results in this paper. Detailed descriptions of the experimental settings can be found in Section 4.1. Implementation details for all the baseline methods and our proposed PASS are elaborated in Section 4.1 and Appendix B. Additionally, the codes are provided in the supplementary materials.

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

## A    PARAMETERS OF HYPERNETWORKS

In this study, the hidden size of the hypernetwork is configured to $64$. A detailed breakdown of the number of parameters for the hypernetworks utilized in this research is provided in Table 3. It is noteworthy that the parameter count for the hypernetworks is significantly lower compared to that of the pretrained models. For instance, in the case of ResNet-18, the hypernetwork parameters constitute only $2.8\%$ of the total parameters of the pre-trained ResNet-18.

Table 3: The number of parameters for our Hypernetworks.

|  | ResNet-18 (11M) | ResNet-34 (21M) | ResNet-50 (25M) | VGG-16 (138M) |
|---|---|---|---|---|
| #Parameters-HyperNetwork | 0.31M (2.8%) | 0.56M (2.6%) | 1.5M (6%) | 0.34M (0.2%) |

## B    IMPLEMENTATION DETAILS

Table 4 summarizes the hyper-parameters for `PASS` used in all our experiments.

Table 4: Implementation details on each dataset.

| Settings | Tny-ImageNet | CIFAR-10 | CIFAF-10 | DTD | StanfordCars | Food101 |
|---|---|---|---|---|---|---|
| *Stage 1: Learning to Prune* | | | | | | |
| Batch Size | | | 128 | | | |
| Weight Decay - VP | 0 | 0 | 0 | 0 | 0 | 0 |
| Learning Rate - VP | $1e-2$ | $1e-2$ | $1e-2$ | $1e-2$ | $1e-2$ | $1e-2$ |
| Optimizer - VP | | | SGD optimizer | | | |
| LR-Decay-Scheduler - VP | | | cosine | | | |
| Weight Decay - HyperNetwork | $1e-2$ | $1e-2$ | $1e-2$ | $1e-2$ | $1e-2$ | $1e-2$ |
| Learning Rate - HyperNetwork | $1e-3$ | $1e-3$ | $1e-3$ | $1e-3$ | $1e-3$ | $1e-3$ |
| Optimizer - HyperNetwork | | | AdamW optimizer | | | |
| LR-Decay-Scheduler - HyperNetwork | | | cosine | | | |
| Total epochs | | | 50 | | | |
| *Stage 2: Fine-tune* | | | | | | |
| Batch Size | | | 128 | | | |
| Weight Decay - VP | 0 | 0 | 0 | 0 | 0 | 0 |
| Learning Rate - VP | $1e-3$ | $1e-2$ | $1e-2$ | $1e-2$ | $1e-2$ | $1e-2$ |
| Optimizer - VP | | | SGD optimizer | | | |
| LR-Decay-Scheduler - VP | | | cosine | | | |
| Weight Decay - Pruned Network | $5e-4$ | $3e-4$ | $5e-4$ | $5e-4$ | $5e-4$ | $5e-4$ |
| Learning Rate - Pruned Network | $1e-3$ | $1e-2$ | $1e-2$ | $1e-2$ | $1e-2$ | $1e-2$ |
| Optimizer - Pruned Network | | | SGD optimizer | | | |
| LR-Decay-Scheduler - Pruned Network | multistep-$\{6, 8\}$ | cosine | cosine | cosine | cosine | cosine |
| Total epochs | 10 | 50 | 50 | 50 | 50 | 50 |

## C    LEARNED CHANNEL SPARSITY

We present the channel sparsity learned by PASS on CIFAR-100 and Tiny-ImageNet using a pre-trained ResNet-18 in Table 5. Our observations indicate that channel sparsity is generally higher in the top layers and lower in the bottom layers of the network.

Table 5: Layer-wise sparsity of the pre-trained ResNet-18 on CIFAR-100 and Tiny-ImageNet as learned by PASS at $30\%, 50\%$ sparsity levels.

| Layer | Fully Dense #Channels | CIFAR-100 | | Tiny-ImageNet | |
|---|---|---|---|---|---|
| | | 30% Sparsity | 50% Sparsity | 30% Sparsity | 50% Sparsity |
| Layer 1 - conv1 | 64 | 9.4% | 29.7% | 20.3% | 37.5% |
| Layer 2 - layer1.0.conv1 | 64 | 17.2% | 43.8% | 28.1% | 56.2% |
| Layer 3 - layer1.0.conv2 | 64 | 29.7% | 29.7% | 20.3% | 39.0% |
| Layer 4 - layer1.1.conv1 | 64 | 15.7% | 46.9% | 50% | 62.5% |
| Layer 5 - layer1.1.conv2 | 64 | 22.7% | 26.6% | 17.1% | 32.8% |
| Layer 6 - layer2.0.conv1 | 128 | 19.6% | 46.9% | 42.9% | 57.0% |
| Layer 7 - layer2.0.conv2 | 128 | 19.6% | 45.4% | 14.0% | 34.3% |
| Layer 8 - layer2.0.downsample.0 | 128 | 19.6% | 45.4% | 14.0% | 34.3% |
| Layer 9 - layer2.1.conv1 | 128 | 19.6% | 44.6% | 34.3% | 55.5% |
| Layer 10 - layer2.1.conv2 | 128 | 7.1% | 28.9% | 16.4% | 34.3% |
| Layer 11 - layer3.0.conv1 | 256 | 29% | 50% | 42.1% | 58.9% |
| Layer 12 - layer3.0.conv2 | 256 | 15.3% | 50% | 11.7% | 28.5% |
| Layer 13 - layer3.0.downsample.0 | 256 | 15.3% | 50% | 11.7% | 28.5% |
| Layer 14 - layer3.1.conv1 | 256 | 27.4% | 43.8% | 33.2% | 52.3% |
| Layer 15 - layer3.1.conv2 | 256 | 11% | 26.2% | 10.1% | 23.8% |
| Layer 16 - layer4.0.conv1 | 512 | 29.3% | 50% | 33.3% | 54.4% |
| Layer 17 - layer4.0.conv2 | 512 | 41.8% | 50% | 36.1% | 58.9% |
| Layer 18 - layer4.0.downsample.0 | 512 | 41.8% | 50% | 36.1% | 58.9% |
| Layer 19 - layer4.1.conv1 | 512 | 44.6% | 48.3% | 39.2% | 65.8% |
| Layer 20 - layer4.1.conv2 | 512 | 42.6% | 49.9% | 27.1% | 46.2% |
| Layer 21 - Linear | 512 | 0% | 0% | 0% | 0% |

## D    EXPERIMENTS ON IMAGENET AND ADVANCED ARCHITECTURES

To draw a solid conclusion, we further conduct extensive experiments on large dataset ImageNet using the advanced pre-trained models such as ResNext-50, Swin-T, and ViT-B/16. The results are shown in Table 6. We observe that our method PASS demonstrates a significant speed-up with minimal accuracy loss, as indicated by the $\Delta$ Acc., which is superior to existing methods like SSS, GFP, and DepGraph. the resulting empirical evidence robustly affirms the effectiveness of PASS across both advanced neural network architectures and large-scale datasets.

Table 6: Pruning results based on ImageNet and Advanced models.

| Arch. | Method | Base | Pruned | $\Delta$ Acc. | FLOPs |
|---|---|---|---|---|---|
| ResNeXt-50 | ResNeXt-50 | 77.62 | - | - | 4.27 |
| | SSS Huang & Wang (2018) | 77.57 | 74.98 | -2.59 | 2.43 |
| | GFP Liu et al. (2021a) | 77.97 | 77.53 | -0.44 | 2.11 |
| | DepGraph Fang et al. (2023) | 77.62 | 76.48 | -1.14 | 2.09 |
| | Ours (PASS) | 77.62 | 77.21 | -0.41 | 2.01 |
| ViT-B/16 | ViT-B/16 | 81.07 | - | - | 17.6 |
| | DepGraph Fang et al. (2023) | 81.07 | 79.17 | -1.90 | 10.4 |
| | Ours(PASS) | 81.07 | 79.77 | -1.30 | 10.7 |
| Swin-T | Swin-T | 81.4 | - | - | 4.49 |
| | X-Pruner Yu & Xiang (2023) | 81.4 | 80.7 | -0.7 | 3.2 |
| | STEP Li et al. (2021) | 81.4 | 77.2 | -4.2 | 3.5 |
| | Ours(PASS) | 81.4 | 80.9 | -0.5 | 3.4 |

# E   COMPLEXITY ANALYSIS OF THE HYPERNETWORK

In this section, we provide a comprehensive analysis about the complexity of the Hypernetwork. (1) Regarding the impact on time complexity, our recurrent hyper-network is designed for efficiency. The channel masks are pre-calculated, eliminating the need for real-time generation during both the inference and subnetwork fine-tuning phases. Therefore, the recurrent hyper-network does not introduce any extra time complexity during the inference and the fintune-tuning phase. The additional computing time is limited to the phase of channel mask identification. (2) Moreover, the hyper-network itself is designed to be lightweight. The number of parameters it contributes to the overall model is minimal, thus ensuring that any additional complexity during the mask-finding phase is negligible. This claim is substantiated by empirical observations: the hyper-network accounts for only about $0.2\%$ to $6\%$ of the total model parameters across various architectures such as ResNet-18/34 and VGG-16, as illustrated in Table 3. (3) Additionally, we assessed the training time per epoch with and without the hyper-network during the channel mask identification phase. Our findings in Table 7 indicate that the inclusion of the LSTM network has a marginal effect on these durations, further affirming the efficiency of our approach.

Table 7: Training Time (s) per Epoch w/ and w/o Hypernetworks during Channel Mask Identification Phase with single A100 GPU.

|  | ResNet-18 (11M) | ResNet-34 (21M) | ResNet-50 (25M) |
| --- | --- | --- | --- |
| w/o HyperNetwork | 70.05 | 73.95 | 95.65 |
| w/ HyperNetwork | 72.2 | 76.95 | 111.6 |

# F   DIFFERENCE BETWEEN OUR PROPOSED PASS AND DYNAMIC NEURAL NETWORK

There are two fundamental differences between our proposed PASS and dynamic neural network. (1) **The hyper-network in our proposed PASS is not 'dynamic'.** Dynamic neural networks, as categorized in the literature, are networks capable of adapting their structures or parameters conditioned in a sample-dependent manner, as outlined in Han et al. (2021). In contrast, the hyper-network within our PASS framework does not exhibit this 'dynamic' nature. It is designed to be dependent on a visual prompt (task-specific), as opposed to dynamically adjusting to input samples. This hyper-network's role is confined to the channel mask identification phase and is not employed during the inference phase. Therefore, it is fundamentally different from dynamic neural networks. (2) **Their goals are different.** The fundamental goal of the hyper-network in PASS is distinct from that of dynamic neural networks. While the latter focuses on adapting their architecture or parameters based on input samples, our hyper-network is specifically engineered for the integration of visual prompts with the statics of model weights.

