# OpenReview forum: "($\texttt{PASS}$) Visual Prompt Locates Good Structure Sparisty through a Recurent HyperNetwork"
_ICLR.cc/2024/Conference — Submitted to ICLR 2024_

### Official Review · Reviewer_dU7f · 2023-10-22

**Soundness:** 2 fair
**Presentation:** 2 fair
**Contribution:** 2 fair
**Rating:** 5
**Confidence:** 4

**Summary:**

The paper introduces a recurrent mechanism with LSTM to acquire layer-wise sparse masks, considering both the sparse masks from previous layers and visual prompts.

The paper has achieved commendable performance on CIFAR and Tiny ImageNet datasets.

**Strengths:**

1. The presentation of this paper is excellent, with professional handling of formulas, images, and expressions.

2. The paper has achieved commendable performance on CIFAR and Tiny ImageNet datasets.

**Weaknesses:**

1. The novelty of this paper is relatively limited. Several methods have already been proposed to address the intricate dependencies arising from channel elimination across layers with sequence network, such as the RNN-based SkipNet[1]. To enhance the novelty, the author is encouraged to explore a broader range of dynamic neural network literature, as numerous ideas and methods have been introduced in this domain over the years.

It is necessary to compare these methods and elucidate their differences.

The reviewer possesses a deep understanding of dynamic networks with sequence modeling. Any potential misconceptions in the reviewer's understanding can be clarified during the rebuttal phase.

2. Visual prompts are typically designed for fine-tuning with limited data and domain transfer scenarios (e.g., transform the ImageNet model to CIFAR), but the author claims that the visual prompt plays a key role in pruning. However, the experiments in this work seem challenging to support this argument, as all the gains from visual prompts appear to be very marginal, less than or equal to 1%. Such experimental results are hard to be convincing.

Additionally, prompt learning relies on a strong foundation of pre-trained models. To demonstrate its effectiveness in network pruning, favorable experiments and analyses are essential.

 In cases where pruning a model without fine-tuning, the visual prompt is unnecessary, in such a scenario, it seems that the paper may not work.

3. The experiments conducted on small datasets, such as CIFAR and Tiny-ImageNet, with very low resolution and data scale are not entirely convincing. The reviewer suggests including experiments on at least ImageNet-1k or ImageNet. In the era of big data, ImageNet is considered a relatively small dataset.

[1] Wang, Xin, Fisher Yu, Zi-Yi Dou, Trevor Darrell, and Joseph E. Gonzalez. "Skipnet: Learning dynamic routing in convolutional networks." In Proceedings of the European Conference on Computer Vision (ECCV), pp. 409-424. 2018.


----------------------

After reading the rebuttal, the reviewer raised the score to 5.

**Questions:**

1. Could the author explain that why the visual prompts improve channel pruning? Since the visual prompts are static across a task or a dataset, why the author state their pruning method as “from a data-centric perspective” while it is not even input dependent?

---

> ### Author Response · Authors · 2023-11-19
> **Response to Reviewer dU7f (1/4)**
>
> **We sincerely appreciate your detailed comments. We are glad that you found our presentation, formulas, images, and expressions professional. We provide point-wise responses to address your concerns below.**
>
> **Q1:** *The novelty of this paper is relatively limited. Several methods have already been proposed to address the intricate dependencies arising from channel elimination across layers with sequence network, such as the RNN-based SkipNet[1]. To enhance the novelty, the author is encouraged to explore a broader range of dynamic neural network literature, as numerous ideas and methods have been introduced in this domain over the years. It is necessary to compare these methods and elucidate their differences.*
>
> **Reply:**
>
> **Cons: Limited novelty:** We respectfully disagree with this assessment and would like to emphasize the technical novelty of our paper, which is primarily manifested in two aspects:
>
> - **Exploring visual prompts in pruning vision models is novel**: As appreciated by **Reviewer F4Cf** and **Reviewer NTwF**, our exploration of visual prompts to network pruning is novel and innovative. Recent advances like in-context learning and prompting have demonstrated great success in the field of LLMs. [1] demonstrates that learning post-pruning prompting can improve the performance of pruned LLMs without changing their weights. However, the effect of visual prompts on sparsifying vision models remains mysterious, due to the fact that visual prompts are inherently more complex to comprehend and typically pose greater challenges in terms of both design and learning, in comparison to their linguistic counterparts. To this end, we propose a novel framework that enabling incorporating visual prompting into the sparsification process, opening the door to the exciting possibility to leveraging visual prompting to prune vision models.
>
> - **The framework for fusing visual prompt and model information is novel**: We introduce a novel framework designed to fuse visual prompt data with model information specifically for channel pruning. The complexity of this challenge stems from the disparate nature of visual prompts and model information, each embodying unique data structures and characteristics. Integrating these two elements effectively for structural pruning is not trivial. Through our thorough ablation studies, we provide empirical evidence showcasing the efficacy of our framework in seamlessly incorporating visual prompts with model information for better structural pruning.
>
> **Cons: Difference between our proposed PASS and dynamic neural network?:** There are two fundamental differences between our proposed PASS and dynamic neural network.
>
> - **The hyper-network in our proposed PASS is not 'dynamic'.** Dynamic neural networks, as categorized in the literature, are networks capable of adapting their structures or parameters conditioned in a sample-dependent manner, as outlined in [2]. In contrast, the hyper-network within our PASS framework does not exhibit this 'dynamic' nature. It is designed to be dependent on a visual prompt (task-specific), as opposed to dynamically adjusting to input samples. This hyper-network's role is confined to the channel mask identification phase and is not employed during the inference phase. Therefore, it is fundamentally different from dynamic neural networks.
>
> - **Their goals are different.** The fundamental goal of the hyper-network in PASS is distinct from that of dynamic neural networks. While the latter focuses on adapting their architecture or parameters based on input samples, our hyper-network is specifically engineered for the integration of visual prompts with the statics of model weights.
>
>
> Additionally, the selection of LSTM as the hyper-network was a deliberate choice, driven by its proven ability to effectively manage inherent channel dependencies via the recurrent mechanism. The efficacy of this choice has been rigorously validated through our ablation studies, as detailed in Section 5.1 (In our submission).
>
> We will update the description for differences of our proposed PASS and dynamic neural network in the revised version (in Appendix F).
>
> [1] Xu, Zhaozhuo, Zirui Liu, Beidi Chen, Yuxin Tang, Jue Wang, Kaixiong Zhou, Xia Hu, and Anshumali Shrivastava. "Compress, Then Prompt: Improving Accuracy-Efficiency Trade-off of LLM Inference with Transferable Prompt." arXiv preprint arXiv:2305.11186 (2023).
>
> [2] Han, Yizeng, et al. "Dynamic neural networks: A survey." IEEE Transactions on Pattern Analysis and Machine Intelligence 44.11 (2021): 7436-7456.

---

> ### Author Response · Authors · 2023-11-19
> **Response to Reviewer dU7f (2/4)**
>
> **Q2:** *Visual prompts are typically designed for fine-tuning with limited data and domain transfer scenarios (e.g., transform the ImageNet model to CIFAR), but the author claims that the visual prompt plays a key role in pruning. However, the experiments in this work seem challenging to support this argument, as all the gains from visual prompts appear to be very marginal, less than or equal to 1%. Such experimental results are hard to be convincing.*
>
> **Reply:** Thank you for your detailed comments.
>
> - **Gains of visual prompt are not ``Marginal":** While the improvements observed with visual prompts in our ablation study (in Section 5.2 and Table 2) are between 0.6% to 1.3%, it is crucial to **contextualize these gains within the domain of network pruning**. In this field, the improvement of 1% is already very appealing, constrained by the limited parameter count. For example, even the current state-of-the-art such as DepGraph[1] and GrowRegularization[2] only outperform their baselines by less than 1%. The gains from visual prompts, although seemingly modest, are non-trivial when considering the challenges associated with further optimizing highly efficient models. Furthermore, these improvements are consistent across different sparsities, indicating a reliable enhancement rather than a statistical anomaly.
>
>
>     **[1]** Fang, Gongfan, et al. "Depgraph: Towards any structural pruning." Proceedings of the IEEE/CVF Conference on Computer Vision and Pattern Recognition. 2023.
>
>     **[2]** Wang, Huan, et al. "Neural pruning via growing regularization." ICLR, (2021).
>
> - **Gains of our overall Framework (PASS) are not ``marginal".** Our proposed PASS demonstrates significant performance enhancements across a range of architectures, datasets, and sparsity levels. Notably, with ResNet-18, our PASS consistently surpasses baseline models by **1% to 3%** in accuracy across six downstream tasks under 1000M FLOPs (from Figure 2 in the submission). Additionally, at a modest speed up, the subnetwork identified by our PASS outperforms fully-finetuned dense networks by **0.5% to 1%** in accuracy, as illustrated in **Table 1**. These results clearly indicate that the benefits of our method are not 'marginal'.
>
>
>     **Table 1: Comparison of Accuracy for PASS and dense neural network under different Datasets**
>
>     | Dataset      | ACC (PASS) | ACC (Dense) | FLOPs (PASS) | FLOPs (Dense) | Δ ACC |
>     |--------------|------------|-------------|--------------|---------------|-------|
>     | CIFAR-10     | 96.77      | 96.02       | 1644M        | 1822M         | 0.75  |
>     | CIFAR-100    | 83.45      | 82.40       | 1647M        | 1822M         | 1.05  |
>     | TinyImageNet | 73.64      | 73.10       | 1491M        | 1822M         | 0.54  |
>     | DTD          | 70.16      | 69.15       | 1677M        | 1822M         | 1.01  |
>     | Food101      | 82.13      | 81.07       | 1700M        | 1822M         | 1.06  |
>
> **Q3:** *Additionally, prompt learning relies on a strong foundation of pre-trained models. To demonstrate its effectiveness in network pruning, favorable experiments and analyses are essential.*
>
> **reply**: Thank you for your insightful feedback. We totally agree with the significance of a strong foundation of pre-trained models for prompt learning. Due to the time limitation, we could not extend to evaluating the effectiveness of VP on our proposed PASS using a very strong foundation of pre-trained vision models such as CLIP[1]. However, we provide a similar analysis to study the effect of model size on the pruning performance caused by visual prompts, by varying the pre-trained model size from ResNet-18 to ResNet-34, and to ResNet-50. The results are shown in **Table 2**.
>
> - The results show an increase in the efficacy of VP as we progress from ResNet-18 to more advanced models like ResNet-34 and ResNet-50. This observation suggests that the impact of VP on the PASS framework might be more pronounced when implemented on stronger foundational vision models.
>
>     **Table 2: Performance of subnetworks (Channel Sparsity: 30%) found by PASS w/ VP and w/o VP with varying different scales of the pre-trained models. The  task is CIFAR-100.**
>     |                | ResNet-18 | ResNet-34 | ResNet-50 |
>     |----------------|-----------|-----------|-----------|
>     | LSTM+Weights   | 81.13     | 82.68     | 82.84     |
>     | LSTM+Weights+VP(PASS) | 81.72 | 83.69 | 83.91 |
>     | Δ              | 0.59      | 1.01      | 1.07      |
>
> [1] Radford, Alec, et al. "Learning transferable visual models from natural language supervision." International conference on machine learning. PMLR, 2021.

---

> ### Author Response · Authors · 2023-11-19
> **Response to Reviewer dU7f (3/4)**
>
> **Q4:** *In cases where pruning a model without fine-tuning, the visual prompt is unnecessary, in such a scenario, it seems that the paper may not work.*
>
> **Reply:** Thank you for your comments. We would like to clarify that our proposed PASS still works in cases where pruning a model without fine-tuning. It is because our proposed PASS concurrently optimizes both the visual prompt and the hypernetwork for finding the channel topology.
>
> - To substantiate our claim, we conducted additional experiments comparing the performance of subnetworks identified by PASS with and without the visual prompt (VP). Our results in **Table 3** demonstrate that subnetworks pruned using PASS with the visual prompt outperform those pruned without it, underscoring the visual prompt's value in structural pruning.
>
>     **Table 3: Performance of subnetworks before fine-tuning. The experiments are based on CIFAR-100 and ResNet-18.**
>
>     |Channel Sparsity | 10%   | 30%   | 50%   | 70%   |
>     |---------------|-------|-------|-------|-------|
>     | LSTM+Weights  | 33.22 | 32.80 | 21.19 | 8.42  |
>     | LSTM+Weights+VP(PASS) | 35.64 | 33.79 | 22.38 | 9.15  |
>
>
> **Q5:** *The experiments conducted on small datasets, such as CIFAR and Tiny-ImageNet, with very low resolution and data scale are not entirely convincing. The reviewer suggests including experiments on at least ImageNet-1k or ImageNet. In the era of big data, ImageNet is considered a relatively small dataset.*
>
> **Reply:** Thank you for your comments. In response, we expanded our empirical investigation to include the ImageNet-1k dataset. Following [5], we adopt the Vision Transformer (ViT-B/16) and ResNeXt-50 architectures. We utilized the pre-trained models from PyTorch Vision ([1]).
>
> - The results are shown in **Table 4**. The results show that our method PASS achieves superior performance over baselines with significant speed-ups with only minimal impacts on accuracy, affirming the effectiveness of our proposed PASS on large datasets.
>
>     **Table 4: Pruning results  on ImageNet with Advanced models.**
>
>     | Arch.      | Method       | Base  | Pruned | Δ Acc. | FLOPs |
>     |------------|--------------|-------|--------|--------|-------|
>     | ResNeXt-50 | ResNeXt-50   | 77.62 | -      | -      | 4.27  |
>     |            | SSS [2]      | 77.57 | 74.98  | -2.59  | 2.43  |
>     |            | GFP [3]      | 77.97 | 77.53  | -0.44  | 2.11  |
>     |            | DepGraph [4] | 77.62 | 76.48  | -1.14  | 2.09  |
>     |            | Ours (PASS)  | 77.62 | 77.21  | -0.41  | 2.01  |
>     | ViT-B/16   | ViT-B/16     | 81.07 | -      | -      | 17.6  |
>     |            | DepGraph [4] | 81.07 | 79.17  | -1.90  | 10.4  |
>     |            | Ours (PASS)  | 81.07 | 79.77  | -1.30  | 10.7  |
>
> [1] [TorchVision](https://pytorch.org/vision/stable/index.html)
>
> [2] Huang, Zehao, and Naiyan Wang. "Data-driven sparse structure selection for deep neural networks." Proceedings of the European conference on computer vision (ECCV). 2018.
>
> [3] Liu, Liyang, et al. "Group fisher pruning for practical network compression." International Conference on Machine Learning. PMLR, 2021.
>
> [4] Fang, Gongfan, et al. "Depgraph: Towards any structural pruning." Proceedings of the IEEE/CVF Conference on Computer Vision and Pattern Recognition. 2023.

---

> ### Author Response · Authors · 2023-11-19
> **Response to Reviewer dU7f (4/4)**
>
> **Q6:** *(1) Could the author explain why the visual prompts improve channel pruning? Since the visual prompts are static across a task or a dataset, why does the author state their pruning method as “from a data-centric perspective” while it is not even input dependent?*
>
> **Reply:** Thank you for your insightful questions. We answer your questions step by step.
>
> **Why visual prompts help channel pruning.**
> - Several studies demonstrated that model sparsification is influenced by the nature of tasks or datasets[1][2]. Visual prompts act as a form of meta-information, encapsulating crucial insights about the task or dataset. It provides the task-dependent information for the pruning process, leading to the identification of more relevant and efficient channel structures. The robust performance of our proposed PASS, demonstrated across six diverse datasets and multiple architectures, further substantiates this claim. Additionally, our ablation studies provide concrete evidence of the efficacy of visual prompts in enhancing channel pruning.
>
> - Furthermore, the increasing volume of research indicates that incorporating information from inputs is crucial for enhancing the performance of pruning algorithms [3][4][5][6]. For example, the recent study [6] highlights that leveraging prompts post-pruning can significantly improve the efficiency of sparse LLMs. Inspired by these insights, we explored how to integrate visual prompts into the pruning process of the vision models and proposed our PASS, showing its superiority over other baselines.
>
> [1] Liu, Shiwei, et al. "Sparsity May Cry: Let Us Fail (Current) Sparse Neural Networks Together!." ICLR (2023).
>
> [2] Yin, Lu, et al. "Junk DNA Hypothesis: A Task-Centric Angle of LLM Pre-trained Weights through Sparsity." arXiv preprint arXiv:2310.02277 (2023).
>
> [3] Sun, Mingjie, et al. "A Simple and Effective Pruning Approach for Large Language Models." arXiv preprint arXiv:2306.11695 (2023).
>
> [4] Frantar, Elias, and Dan Alistarh. "SparseGPT: Massive Language Models Can Be Accurately Pruned in One-Shot." ICML (2023).
>
> [5] Yin, Lu, et al. "Outlier Weighed Layerwise Sparsity (OWL): A Missing Secret Sauce for Pruning LLMs to High Sparsity." arXiv preprint arXiv:2310.05175 (2023).
>
> [6] PROMPT, TRANSFERABLE. "Compress, Then Prompt: Improving Accuracy-Efficiency Trade-off of LLM Inference with Transferable Prompt." arXiv preprint arXiv:2305.11186 (2023).
>
> **Our proposed PASS is from a data-centric perspective.**
>
> - Firstly, let us elucidate the concept of a data-centric technique. A data-centric technique **does not necessarily hinge on being input sample-dependent**. Instead, it can also be task-specific or dataset-dependent[1]. This kind of technique primarily focuses on engineering and managing data to boost model performance. A notable manifestation of this in the data-centric paradigm is the utilization of prompts, including visual prompts, which are specific to a task or dataset[1]. In our proposed PASS, the reliance on visual prompts categorically aligns it with a data-centric methodology, as these prompts are dataset-specific. Therefore, our method's designation as data-centric is aptly justified.
>
> [1] Zha, Daochen, et al. "Data-centric artificial intelligence: A survey." arXiv preprint arXiv:2303.10158 (2023).

---

> > ### Comment · Reviewer_dU7f · 2023-11-21
> > **Response to authors**
> >
> > Thank you for the author's response. The author has addressed most of my concerns.
> >
> > However, the reviewer still expresses concerns about novelty. Although this work focuses on a different task and has a different objective compared to dynamic networks, the network architecture appears to be similar.
> >
> > Additionally, in Table 4: Pruning results on ImageNet with Advanced models, compared to previous works, the gain in this study appears to be marginal, with an increase of approximately 0.6%.
> >
> > Furthermore, the authors only provided experiments for ResNeXt-50 with 50% sparsity and experiments for ViT-B/16 with around 40% sparsity. Although the reviewer acknowledges the challenges of conducting additional experiments in a short time frame, suggesting that the authors provide results for more challenging and higher sparsity settings, such as 60% or 70%.
> >
> > In summary, the reviewer gives a final score of 5: marginally below the acceptance threshold.

---

> > > ### Author Response · Authors · 2023-11-21
> > > **Response to Reviewer dU7f**
> > >
> > > **Q1:** *However, the reviewer still expresses concerns about novelty. Although this work focuses on a different task and has a different objective compared to dynamic networks, the network architecture appears to be similar.*
> > >
> > > **Reply:** Thank you for recognizing the distinction in our research area from dynamic networks. Our focus differs significantly as we aim to identify which channels can be permanently pruned for all inputs, rather than dynamically determining which layer to skip for each input sample. This task poses a greater challenge as the removed channels are eliminated permanently for all inputs without reinstatement during the inference process.
> > >
> > > Furthermore, we want to highlight our **primary novelty**: the pioneering integration of visual prompts in the realm of structured pruning. We for the first time introduce visual prompts into pruning and designing a recurrent hypernetwork to merge visual prompts with model information, thereby advancing structured pruning techniques.
> > >
> > > All of the above points are outside of the scope of the dynamic neural network literature. **We are confident that our paper makes a novel and significant contribution to the domain of pruning.**
> > >
> > > **Q2:** *Additionally, in Table 4: Pruning results on ImageNet with Advanced models, compared to previous works, the gain in this study appears to be marginal, with an increase of approximately 0.6\%.*
> > >
> > > **Reply:** We want to highlight that 0.6% accuracy improvement over the state-of-the-art (SOTA) method is already significant in the field of structured pruning:
> > >
> > > - It is crucial to contextualize these gains within the framework of ImageNet's already saturated performance levels. In the realm of state-of-the-art methods, such as DepGraph with ViT-B/16, the performance gap between pruned subnetworks and dense networks is notably narrow (only 1.9%). Thus, our improvement of 0.6% over DepGraph should be viewed as a significant achievement, given this saturation.
> > >
> > > - Our improvements, which include a maximum gain of up to 3%, are consistent across seven datasets, ranging from small to large datasets, and on various architectures from convolution-based to transformer-based models.
> > >
> > > - In addition, the channel mask found by our method demonstrates superior transferability across other datasets, demonstrating its broad generalization capability among various subsequent tasks.
> > >
> > > [1] Fang, Gongfan, et al. "DepGraph: Towards Any Structural Pruning." Proceedings of the IEEE/CVF Conference on Computer Vision and Pattern Recognition. 2023.
> > >
> > > **Q3:** *Furthermore, the authors only provided experiments for ResNeXt-50 with 50\% sparsity and experiments for ViT-B/16 with around 40\% sparsity. Although the reviewer acknowledges the challenges of conducting additional experiments in a short time frame, suggesting that the authors provide results for more challenging and higher sparsity settings, such as 60\% or 70\%.*
> > >
> > > **Reply:** Thanks for your feedback.
> > >
> > > - We follow the state-of-the-art structured pruning studies - DepGraph[1], Xpruner[2], GFP[3] - and prune our model to around 50% FLOPs as the dense model. We clarify that in the context of structured pruning, a 50% reduction in FLOPs is a much more practical setting, offering considerable speedup while maintaining comparable performance. Pruning with 60% and 70% sparsity would lead to significant performance degradation, making the performance level too poor for practical use.
> > >
> > > - Still, as you requested, we have launched experiments with 60% and 70% sparsity levels, whose estimated running time is two and a half days. Given the limited rebuttal time window (40 hours left until the rebuttal deadline), it is not feasible for us to complete these experiments within this period. We will update them in our next version once they are available.
> > >
> > > - Nevertheless, we have provided results of 60% FLOPs reduction on a small-scale ImageNet dataset — TinyImageNet — in Figure 2 of our submission, where our approach consistently outperforms state-of-the-art baselines, demonstrating its efficacy even at higher levels of sparsity.
> > >
> > > [1] Fang, Gongfan, et al. "DepGraph: Towards Any Structural Pruning." Proceedings of the IEEE/CVF Conference on Computer Vision and Pattern Recognition. 2023.
> > >
> > > [2] Yu, Lu, and Wei Xiang. "X-Pruner: eXplainable Pruning for Vision Transformers." Proceedings of the IEEE/CVF Conference on Computer Vision and Pattern Recognition. 2023.
> > >
> > > [3] Liu, Liyang, et al. "Group Fisher Pruning for Practical Network Compression." International Conference on Machine Learning. PMLR, 2021.

---

### Official Review · Reviewer_dN4F · 2023-10-30

**Soundness:** 3 good
**Presentation:** 3 good
**Contribution:** 3 good
**Rating:** 5
**Confidence:** 4

**Summary:**

In this paper, the authors study how to use visual prompts for channel pruning. The authors argue that the layer-wise mask should consider the sequential dependency between adjacency layers, network weights and visual prompts. Motivated by this argument, the authors propose PASS to learn sparse mask using a recurrent LSTM network. The authors conduct experiments on six target datasets with four different backbones.

**Strengths:**

1. The proposed method achieves better performance over the baselines on most of the proposed settings.
2. The authors provide the code in the appendix.

**Weaknesses:**

1. Model Complexity: While the channel pruning reduces size, the added LSTM network introduces new parameters. An analysis of its impact on model parameters considering the LSTM and training/testing time would be beneficial.
2. Backbone Networks: this paper uses ResNet and VGG as the backbone networks. I recommend the authors also explore more contemporary and potentially powerful architectures, such as ResNeXT and ViT used in DepGraph.
3. Benchmarks: The paper's benchmarks are limited in size. Testing on larger datasets like ImageNet, as used in GrowReg, DepGraph, and other baselines, is recommended.

**Questions:**

1. Please correct the typos in the title, "sparsity" and "recurrent".
2. It's preferable to place figures and tables at the top of a page.
3. The authors may consider switching the sequence of Figure 4 and Figure 5 to align with their respective mentions in the text.

---

> ### Author Response · Authors · 2023-11-19
> **Response to Reviewer dN4F (1/2)**
>
> **We sincerely appreciate your constructive suggestions and detailed comments. We provide point-wise responses to your concerns below.**
>
> **Q1:** *Model Complexity: While the channel pruning reduces size, the added LSTM network introduces new parameters. An analysis of its impact on model parameters considering the LSTM and training/testing time would be beneficial?*
>
> **Reply:** Thank you for your insightful suggestion regarding the analysis of model complexity in relation to the LSTM network used in our approach. We appreciate the emphasis on understanding the overall impact on model parameters and time efficiency during training and testing. In response, we have conducted a comprehensive analysis, the details of which we will incorporate into our revised manuscript (in Appendix E).
>
> - **Regarding the impact on time complexity:** Our recurrent hyper-network is designed for efficiency. The channel masks are pre-calculated, eliminating the need for real-time generation during both the inference and subnetwork fine-tuning phases. Therefore, the recurrent hyper-network does not introduce any extra time complexity during the inference and the fine-tuning phase. The additional computing time is limited to the phase of channel mask identification.
>
> - **Moreover, the hyper-network itself is designed to be lightweight:** The number of parameters it contributes to the overall model is minimal, thus ensuring that any additional complexity during the mask-finding phase is negligible. This claim is substantiated by empirical observations: the hyper-network accounts for only about **0.2% to 6%** of the total model parameters across various architectures such as ResNet-18/34 and VGG-16, as illustrated in **Table 1**.
>
>     **Table 1: The number of parameters for our Hypernetworks.**
>
>     |                           | ResNet-18 (11M) | ResNet-34 (21M) | ResNet-50 (25M) | VGG-16 (138M) |
>     |---------------------------|-----------------|-----------------|-----------------|---------------|
>     | #Parameters-HyperNetwork  | 0.31M (2.8%)    | 0.56M (2.6%)    | 1.5M (6%)       | 0.34M (0.2%)  |
>
> - Additionally, we assessed the training time per epoch with and without the hyper-network during the channel mask identification phase. Our findings in **Table 2** indicate that the inclusion of the LSTM network has a marginal effect on these durations, further affirming the efficiency of our approach.
>
>     **Table 2: Training Time (s) per Epoch w/ and w/o Hypernetworks during Channel Mask Identification Phase with single A100 GPU.**
>
>     |                  | ResNet-18 (11M) | ResNet-34 (21M) | ResNet-50 (25M) |
>     |------------------|-----------------|-----------------|-----------------|
>     | w/o HyperNetwork | 70.05           | 73.95           | 95.65           |
>     | w/ HyperNetwork  | 72.2            | 76.95           | 111.6           |

---

> ### Author Response · Authors · 2023-11-19
> **Response to Reviewer dN4F (2/2)**
>
> **Q2:** *Backbone Networks: this paper uses ResNet and VGG as the backbone networks. I recommend the authors also explore more contemporary and potentially powerful architectures, such as ResNeXT and ViT used in DepGraph.*  **Q3:** *Benchmarks: The paper's benchmarks are limited in size. Testing on larger datasets like ImageNet, as used in GrowReg, DepGraph, and other baselines, is recommended.*
>
> **Reply:** We appreciate your suggestion to explore more contemporary architectures and larger datasets.
>
> - In response, we have extended our experiments to include the Vision Transformer (ViT-B/16) and ResNeXt-50 on ImageNet. We utilized pre-trained models from PyTorch Vision ([1]), ensuring that our experiments are replicable and grounded in standard benchmarks. The results are reported in **Table 3**.
>
> - Notably, our method PASS demonstrates a significant speed-up with minimal accuracy loss, as indicated by the $\Delta$Acc., which are superior to existing methods like SSS, GFP, and DepGraph. the resulting empirical evidence robustly affirms the effectiveness of PASS across both advanced neural network architectures and large-scale datasets.
>
>     **Table 3: Pruning results based on ImageNet and Advanced models.**
>
>     | Arch.      | Method      | Base  | Pruned | Δ Acc. | FLOPs |
>     |------------|-------------|-------|--------|--------|-------|
>     | ResNeXt-50 | ResNeXt-50  | 77.62 | -      | -      | 4.27  |
>     |            | SSS [2]     | 77.57 | 74.98  | -2.59  | 2.43  |
>     |            | GFP [3]     | 77.97 | 77.53  | -0.44  | 2.11  |
>     |            | DepGraph [4]| 77.62 | 76.48  | -1.14  | 2.09  |
>     |            | Ours (PASS) | 77.62 | 77.21  | -0.41  | 2.01  |
>     | ViT-B/16   | ViT-B/16    | 81.07 | -      | -      | 17.6  |
>     |            | DepGraph [4]| 81.07 | 79.17  | -1.90  | 10.4  |
>     |            | Ours(PASS)  | 81.07 | 79.77  | -1.30  | 10.7  |
>
> [1] [TorchVision - Stable](https://pytorch.org/vision/stable/index.html)
>
> [2] Huang, Zehao, and Naiyan Wang. "Data-driven sparse structure selection for deep neural networks." Proceedings of the European conference on computer vision (ECCV). 2018.
>
> [3] Liu, Liyang, et al. "Group fisher pruning for practical network compression." International Conference on Machine Learning. PMLR, 2021.
>
> [4] Fang, Gongfan, et al. "Depgraph: Towards any structural pruning." Proceedings of the IEEE/CVF Conference on Computer Vision and Pattern Recognition. 2023.
>
>
> **Q4:** *Please correct the typos in the title, "sparsity" and "recurrent". It's preferable to place figures and tables at the top of a page.*
>
> **Reply:** Thank you very much for highlighting these important details.
>
> - We have carefully reviewed the manuscript and corrected the typos in the title.
>
> - Additionally, in line with your suggestion for improved readability and standard formatting practices, we have adjusted the layout to position figures and tables at the top of the pages in our revision.
>
> **Q5:** *The authors may consider switching the sequence of Figure 4 and Figure 5 to align with their respective mentions in the text.*
>
> **Reply:** Thank you so much for pointing it out. Following your helpful suggestion, we have rearranged these figures in the revised version.
>
> **If you have any other suggestiongs, please let us know. We are more than happy to address them.**

---

> ### Author Response · Authors · 2023-11-21
> **Last 40 hours reminder**
>
> Dear Reviewer **dN4F**,
>
> Thanks a lot for your constructive reviews! We really hope to have a further discussion with you to see if our response solves your concerns.
>
> In our response, we have **(1)** provided a comprehensive analysis of the extra model complexity introduced by the recurrent hypernetwork. **(2)** presented experimental results with **ImageNet** and advanced architectures such as **ViT-B/16 and ResNext-50**. **(3)** corrected the typos and rearranged the tables and figures as you requested.
>
> With only 40 hours remaining until the ICLR rebuttal deadline, we sincerely hope you can review our response and look forward to your further feedback! Your feedback and support is very important to us! We greatly appreciate that!
>
> Best wishes,
>
> The authors

---

> ### Author Response · Authors · 2023-11-22
> **A friendly reminder**
>
> Dear Reviewer **dN4F**,
>
> This is a friendly reminder that there are only 12 hours left before the Author-Reviewer Discussion Period closes. We are more than happy to provide more information if you still have any concerns. We are looking forward to your further feedback!
>
> Thank you again for your time spent on our rebuttal.
>
> Best regards,
>
> The authors

---

### Official Review · Reviewer_NTwF · 2023-10-31

**Soundness:** 3 good
**Presentation:** 3 good
**Contribution:** 3 good
**Rating:** 8
**Confidence:** 2

**Summary:**

This paper aims to solve the problem of estimating the channel significance in structural pruning task. It leverages the visual prompts in in-context learning to capture the channel significance and derive high-quality structural sparsity. A novel network which takes visual prompts and weight statistics as input will output layer-wise channel sparsity recurrently. Experiments have demonstrated effectiveness of proposed method.

**Strengths:**

1. It is novel to take the visual prompts into the channel pruning problem.
2. The theoretical analysis is solid and convincing for me.
3. Experimental results are sufficient to demonstrate the effectiveness of proposed method.

**Weaknesses:**

No obvious weakness for me.

**Questions:**

Is the proposed method effective to vision transformer based models?

---

> ### Author Response · Authors · 2023-11-19
> **Response to Reviewer NTwF**
>
> **We sincerely appreciate your detailed comments and positive ranking. We provide point-wise responses to your concerns below.**
>
> **Q1:** *Is the proposed method effective for vision transformer-based models?*
>
> **Reply:** Thanks for your concern in transformer-based architecture. We report the performance of PASS on the ViT-B-16 and Swin-T models on ImageNet where the pre-trained ViT-B-16 and Swin-T are from torchvision [1]. The results in **Table 1** show that our method PASS achieves higher accuracy than baselines with similar FLOPs, validating PASS is a general method that can be applied to various architectures.
>
> **Table 1: Pruning results Based on vision transformers on ImageNet.**
>
> | Arch.    | Method      | Base  | Pruned | Δ Acc. | FLOPs |
> |----------|-------------|-------|--------|--------|-------|
> | ViT-B/16 | ViT-B/16    | 81.07 | -      | -      | 17.6  |
> |          | DepGraph [2]| 81.07 | 79.17  | -1.90  | 10.4  |
> |          | Ours (PASS) | 81.07 | 79.77  | -1.30  | 10.7  |
> | Swin-T   | Swin-T      | 81.4  | -      | -      | 4.49  |
> |          | X-Pruner [3]| 81.4  | 80.7   | -0.7   | 3.2   |
> |          | STEP [4]    | 81.4  | 77.2   | -4.2   | 3.5   |
> |          | Ours (PASS) | 81.4  | 80.9   | -0.5   | 3.4   |
>
> [1] [TorchVision - Stable](https://pytorch.org/vision/stable/index.html)
>
> [2] Fang, Gongfan, et al. "Depgraph: Towards any structural pruning." Proceedings of the IEEE/CVF Conference on Computer Vision and Pattern Recognition. 2023.
>
> [3] Yu, Lu, and Wei Xiang. "X-Pruner: eXplainable Pruning for Vision Transformers." Proceedings of the IEEE/CVF Conference on Computer Vision and Pattern Recognition. 2023.
>
> [4] Li, Jiaoda, Ryan Cotterell, and Mrinmaya Sachan. "Differentiable subset pruning of transformer heads." Transactions of the Association for Computational Linguistics 9 (2021): 1442-1459.

---

> > ### Comment · Reviewer_NTwF · 2023-11-22
> > **Response to authors**
> >
> > Thank you for your response. Your response have addressed my concerns and I will keep my score.

---

> > > ### Author Response · Authors · 2023-11-22
> > > **Thanks for your support**
> > >
> > > Dear Reviewer NTwF,
> > >
> > > We sincerely appreciate all the helpful feedback and very positive evaluations from reviewer NTwF.
> > >
> > > Your support means a lot to us. We really appreciate it!
> > >
> > > Warm regards,
> > >
> > > The Authors

---

### Official Review · Reviewer_F4Cf · 2023-11-01

**Soundness:** 2 fair
**Presentation:** 3 good
**Contribution:** 2 fair
**Rating:** 5
**Confidence:** 3

**Summary:**

The paper addresses the inefficiency of large-scale neural networks by proposing a pruning method named PASS, which stands for Prune to Achieve Sparse Structures.PASS utilizes visual prompts as an innovative means to identify crucial channels for pruning, seeking to enhance model efficiency without sacrificing performance. This framework adopts a recurrent hypernetwork to generate sparse channel masks in an auto-regressive manner, leveraging both visual prompts and weight statistics of the network. The authors provide extensive experimental evidence showing that PASS achieves better accuracy with fewer computational resources across multiple datasets and network architectures. They also highlight that the hypernetwork and sparse channel masks generated by PASS have superior transferability for subsequent tasks.

**Strengths:**

1. PASS introduces a novel use of visual prompts to determine channel importance.
2. Using recurrent hyper networks allows efficient learning of sparse masks, considering the inter-layer dependencies.
3. Experiment results show the advantage of the proposed method over baselines on convolution baseline over small benchmarks.

**Weaknesses:**

1. The recurrent hyper network approach might introduce complexity, especially in the LSTM network. Does the FLOPs computation involve the hyper-network? This requires more clear explanation in the paper.
2. This paper only experiments with the convolution-based method. While the transformer-based approach, such as vision transformers and swin-transformers, has no investigations. To validate the generalization of the proposed approach, the authors need to provide more experiments on transformer-based networks.
3. The experiments performed in small-scale datasets, such as cifar10, cifar100. It is worth reporting results on large datasets such as imagenet.

**Questions:**

Please refer to the questions in the weakness section.

**Details Of Ethics Concerns:**

No concern on Ethics.

---

> ### Author Response · Authors · 2023-11-19
> **Response to Reviewer F4Cf (1/2)**
>
> **We sincerely appreciate your detailed comments. We provide point-wise responses to your concerns below.**
>
> **Q1:** *The recurrent hyper-network approach might introduce complexity, especially in the LSTM network. Does the FLOPs computation involve the hyper-network? This requires more clear explanation in the paper.*
>
> **Reply:** Thank you for your helpful feedback. We give detailed explanations here and will incorporate this detailed explanation into the revised version of our paper (in Appendix E).
>
> - We acknowledge the concern about potential complexities arising from the incorporation of an LSTM network. It's crucial to emphasize that our paper primarily **focuses on achieving inference speedup**. The role of the hyper-network is solely dedicated to generating sparse masks. After that, **it will not be used during both the inference and fine-tuning phases**. Therefore, the FLOPs computation does not involve the hyper-network.
>
> - Moreover, the hyper-network itself is designed to be lightweight. The number of parameters it contributes to the overall model is minimal, thus ensuring that any additional complexity during the mask finding phase is negligible. This claim is substantiated by empirical observations: the hyper-network accounts for only about **0.2% to 6%** of the total model parameters across various architectures such as ResNet-18/34/50 and VGG-16, as illustrated in **Table 1**.
>
>     **Table 1:** The number of parameters for our Hypernetworks.
>
>     |                           | ResNet-18 (11M) | ResNet-34 (21M) | ResNet-50 (25M) | VGG-16 (138M)  |
>     |---------------------------|-----------------|-----------------|-----------------|----------------|
>     | #Parameters-HyperNetwork  | 0.31M (2.8%)    | 0.56M (2.6%)    | 1.5M (6%)       | 0.34M (0.2%)   |
>
> - Additionally, we assessed the training time per epoch with and without the hyper-network during the channel mask identification phase. Our findings in Table 2 indicate that the inclusion of the LSTM network has a marginal effect on these durations, further affirming the efficiency of our approach.
>
>     **Table 2:** Training Time (s) per Epoch w/ and w/o Hypernetworks during Channel Mask Identification Phase with single A100 GPU.
>
>     |                  | ResNet-18 (11M) | ResNet-34 (21M) | ResNet-50 (25M) |
>     |------------------|-----------------|-----------------|-----------------|
>     | w/o HyperNetwork | 70.05           | 73.95           | 95.65           |
>     | w/ HyperNetwork  | 72.2            | 76.95           | 111.6           |

---

> ### Author Response · Authors · 2023-11-19
> **Response to Reviewer F4Cf (2/2)**
>
> **Q2:** *This paper only experiments with the convolution-based method. While the transformer-based approach, such as vision transformers and swin-transformers, has no investigations. To validate the generalization of the proposed approach, the authors need to provide more experiments on transformer-based networks.* **Q3:** *The experiments performed in small-scale datasets, such as CIFAR-10, CIFAR-100. It is worth reporting results on large datasets such as ImageNet.*
>
> **Reply:** Thank you for your valuable suggestions emphasizing the need for broader experimentation to validate the generalizability of our proposed PASS methodology.
>
> - In line with your recommendations, we have explored our approach to transformer-based networks and larger dataset, i.e., Vision Transformers (ViT-Base-16) and Swin Transformers (Swin-Tiny) on ImageNet-1K. The pre-trained weights are obtained from torchvision [1].
>
> - The results from these extended experiments, as shown in **Table 3**, illustrate that PASS achieves higher accuracies than baselines with similar FLOPs on ImageNet with various advanced architectures, affirming that our proposed PASS can be generalized to advanced architectures and large datasets.
>
> **Table 3: Pruning results based on ImageNet and Advanced models.**
>
> | Arch.      | Method       | Base  | Pruned | Δ Acc. | FLOPs |
> |------------|--------------|-------|--------|--------|-------|
> | ResNeXt-50 | ResNeXt-50   | 77.62 | -      | -      | 4.27  |
> |            | SSS [2]      | 77.57 | 74.98  | -2.59  | 2.43  |
> |            | GFP [3]      | 77.97 | 77.53  | -0.44  | 2.11  |
> |            | DepGraph [4] | 77.62 | 76.48  | -1.14  | 2.09  |
> |            | Ours (PASS)  | 77.62 | 77.21  | -0.41  | 2.01  |
> | ViT-B/16   | ViT-B/16     | 81.07 | -      | -      | 17.6  |
> |            | DepGraph [4] | 81.07 | 79.17  | -1.90  | 10.4  |
> |            | Ours (PASS)  | 81.07 | 79.77  | -1.30  | 10.7  |
> | Swin-T     | Swin-T       | 81.4  | -      | -      | 4.49  |
> |            | X-Pruner [5] | 81.4  | 80.7   | -0.7   | 3.2   |
> |            | STEP [6]     | 81.4  | 77.2   | -4.2   | 3.5   |
> |            | Ours (PASS)  | 81.4  | 80.9   | -0.5   | 3.4   |
>
>
>
> [1] [TorchVision - Stable](https://pytorch.org/vision/stable/index.html)
>
> [2] Huang, Zehao, and Naiyan Wang. "Data-driven sparse structure selection for deep neural networks." Proceedings of the European conference on computer vision (ECCV). 2018.
>
> [3] Liu, Liyang, et al. "Group fisher pruning for practical network compression." International Conference on Machine Learning. PMLR, 2021.
>
> [4] Fang, Gongfan, et al. "Depgraph: Towards any structural pruning." Proceedings of the IEEE/CVF Conference on Computer Vision and Pattern Recognition. 2023.
>
> [5] Yu, Lu, and Wei Xiang. "X-Pruner: eXplainable Pruning for Vision Transformers." Proceedings of the IEEE/CVF Conference on Computer Vision and Pattern Recognition. 2023.
>
> [6] Li, Jiaoda, Ryan Cotterell, and Mrinmaya Sachan. "Differentiable subset pruning of transformer heads." Transactions of the Association for Computational Linguistics 9 (2021): 1442-1459.

---

> ### Author Response · Authors · 2023-11-21
> **Last 40 hours reminder**
>
> Dear Reviewer **F4Cf**,
>
> Thanks a lot for your constructive reviews! We really hope to have a further discussion with you to see if our response solves your concerns.
>
> In our response, we have provided **(1)** a  comprehensive analysis of the model complexity introduced by the recurrent hypernetwork.  **(2)**  experimental results with ImageNet and transformer-based models.
>
> With only 40 hours remaining until the ICLR rebuttal deadline, we sincerely hope you can review our response and look forward to your further feedback! Your feedback and support is very important to us! We greatly appreciate that!
>
> Best wishes,
>
> The authors

---

> ### Author Response · Authors · 2023-11-22
> **A friendly reminder**
>
> Dear Reviewer **F4Cf**,
>
> This is a friendly reminder that there are only 12 hours left before the Author-Reviewer Discussion Period closes.  We are more than happy to provide more information if you still have any concerns. We are looking forward to your further feedback!
>
> Thank you again for your time spent on our rebuttal.
>
> Best regards,
>
> The authors

---

### Meta-Review · Area_Chair_oAi8 · 2023-12-12

**Metareview:**

**Summary**:

The paper presents a network pruning technique that extends and utilizes visual prompts from in-context learning to identify informative channels, resulting in high-quality structural sparsity. This approach thus enhances model efficiency while aiming to preserve performance. The core idea is to employ a recurrent hypernetwork that generates layer-wise channel sparsity in an auto-regressive manner, utilizing both visual prompts and weight statistics of the network. Extensive experimental results demonstrate that the proposed approach achieves improved accuracy with reduced computational resources across various datasets and network backbones. Additionally, the proposed approach demonstrates superior transferability of the hypernetwork and sparse channel masks generated for subsequent tasks.

**Strengths And Weaknesses**:

The reviewers recognize the novelty (introducing visual prompts to determine channel importance), effectiveness, and good results of this work.

In the original paper, the empirical evaluation focused on small-scale datasets such as CIFAR and convolution-based networks. Reviewers raised concerns about the generalizability of the proposed approach to large-scale datasets like ImageNet and transformer-based networks. Additionally, concerns were expressed about the potential increase in model complexity with the introduction of the recurrent hypernetwork. Other points of contention included the perceived limited novelty in comparison to dynamic neural networks and the marginal performance gains from visual prompts.

**Recommendation**:

The authors provided additional results and clarifications in the rebuttal to address the raised concerns. Most reviewers did not provide feedback on the authors' rebuttal.
After checking the rebuttal in detail, the area chairs believe that the primary concern shared by all three negative reviewers remains, which centers around the empirical results regarding larger architecture (e.g., ViT-B) combined with a larger dataset (e.g., ImageNet). More specially,

1) The comparison with prior methods in Table 3 (pruning results based on ImageNet and advanced models) in response to Reviewer F4Cf, and similarly in other responses, appears to be incomplete. Notably, strong baselines such as DepGraph, GFP, and X-Pruner do not appear in every architecture. While it is acknowledged that this omission might be a result of limited rebuttal time, it is important to address for a comprehensive evaluation.

2) The proposed method does not outperform prior methods in terms of both pruned accuracy and FLOPs. In other words, using prior methods can lead to either higher accuracy or less computation. This observation does not support further application of the proposed method.

Overall, while the use of visual prompts in this context is somewhat novel, the reviewers do not identify additional benefits of such an approach. Moreover, the current results on ImageNet cannot strongly support application of the proposed method in real-world settings. The area chairs therefore cannot endorse the acceptance of this work.

**Justification For Why Not Higher Score:**

Although most of the reviewers did not provide feedback on the authors' rebuttal, the area chairs, after checking the rebuttal in detail, believe that the major concern shared by all three negative reviewers remains. While the use of visual prompts in this context is somewhat novel, the reviewers do not identify additional benefits of such an approach. Moreover, the current results on ImageNet cannot strongly support application of the proposed method in real-world settings.

**Justification For Why Not Lower Score:**

N/A

---

### Decision · Program_Chairs · 2024-01-16

Reject